# Engineering Inorganic Pyrophosphate Metabolism as a Strategy to Generate a Fluoride-Resistant *Saccharomyces cerevisiae* Strain

**DOI:** 10.3390/microorganisms13020226

**Published:** 2025-01-21

**Authors:** José R. Perez-Castiñeira, Francisco J. Ávila-Oliva, Aurelio Serrano

**Affiliations:** 1Instituto de Bioquímica Vegetal y Fotosíntesis, Universidad de Sevilla-CSIC, Av. Américo Vespucio 49, 41092 Sevilla, Spain; aurelio@ibvf.csic.es; 2Aplive, 41703 Dos Hermanas, Spain; favilaoliva@hotmail.com

**Keywords:** fluoride resistance, inorganic pyrophosphatase, overexpression, *Saccharomyces cerevisiae*, *Streptococcus mutans*

## Abstract

Fluorine accounts for 0.3 g/kg of the Earth’s crust, being widely distributed in the environment as fluoride. The toxic effects of this anion in humans and other organisms have been known for a long time. Fluoride has been reported to alter several cellular processes although the mechanisms involved are largely unknown. Inorganic pyrophosphatases (PPases) are ubiquitous enzymes that hydrolyze inorganic pyrophosphate (PPi), a metabolite generated from ATP. In *Saccharomyces cerevisiae*, the enzyme responsible for PPi hydrolysis in the cytosol (IPP1) is strongly inhibited by fluoride in vitro. The essentiality of IPP1 for growth has been previously demonstrated using YPC3, a yeast mutant with conditional expression of the corresponding gene. Here, YPC3 was used to generate cells that tolerate high concentrations of fluoride by (a) the overexpression of IPP1 or its human ortholog, or (b) the substitution of IPP1 by the fluoride-insensitive PPase from *Streptococcus mutans*. The results obtained suggest that maintaining appropriate levels of PPase activity in the cytosol is essential for the adaptation of *S. cerevisiae* to high fluoride concentrations. The increase in fluoride tolerance allows YPC3 cells transformed with suitable plasmids to be selected on rich non-selective medium supplemented with this anion.

## 1. Introduction

Fluoride, the smallest of the halide anions, has the highest electronegativity of all the elements of the Periodic Table. It is very abundant in the Earth’s crust (an average of 0.32 g/kg), being found in soils, water and the air. Fluoride concentrations range from 25 μM to 100 mM in water resources, and it may be the most abundant anion in certain groundwaters [1,2]. Consumption of water with a fluoride content close to 0.2 mM can produce dental fluorosis in children, while skeletal fluorosis, a more severe disease, may result from regularly drinking water with higher levels of fluoride (0.2 to 0.8 mM). Skeletal fluorosis causes back pain and rigidity as well as neurological disorders, being endemic in at least 25 countries including India, Mexico, Kenya and Tanzania. The number of people suffering from this disease is estimated to be in the range of tens of millions [3]. Fluoride also exerts effects on other organisms such as bacteria and fungi. The antimicrobial activity of this anion has been studied in recent decades, mainly on bacteria causing dental caries, and it has been reported to be toxic to yeast and other fungi, some of them pathogenic [4].

Due to its ubiquity in the environment and its toxic effects, many organisms have developed strategies of fluoride resistance, although little is known about these strategies and the biochemical pathways involved. Fluoride-responsive riboswitches that regulate the expression of genes in response to this anion have been identified in bacteria and archaea, fluoride exporters being the most common proteins found in operons with these riboswitches [5,6]. Two different fluoride export proteins have been identified in prokaryotes: (a) CLC^F^s, fluoride/proton antiporters, which harness the proton gradient to expel fluoride from the cytoplasm, and (b) Fluc proteins (fluoride channels), passive channels that drive fluoride extrusion down its electrochemical gradient across the plasma membrane. Among the eukaryotes, fungi (including yeast), plants, and some ocean-dwelling animals possess fluoride exporters that belong to a third protein family known as FEX, structurally related to the Flucs [6,7]. Besides extrusion systems, alternative factors have been reported to influence fluoride tolerance [6,8].

Fluoride is known to be an inhibitor of certain major intracellular proteins, and its effects are believed to be exerted by charge interaction with cations such as Mg^2+^ and/or Ca^2+^. In the case of enolase, structural data suggest that the inhibition is due to the formation of a magnesium-fluoride–phosphate complex [9]. Fluoride also has the capacity to form complexes with aluminum and beryllium cations that may act as phosphate emulators, thus inhibiting enzymes involved in the transfer of phosphoryl groups [10]. This large class of enzymes include the so-called inorganic pyrophosphatases (PPases, EC 3.6.1.1), which hydrolyze inorganic pyrophosphate (PPi) [11]. PPi is a by-product of many anabolic reactions, and its efficient removal allows their shift toward biosynthesis according to the law of mass action [12]; as a consequence, accumulation of PPi can collapse anabolism in cellular compartments like cytosol or mitochondria. Two main classes of structurally different PPases have been identified to date: (a) soluble PPases (sPPases), ubiquitous proteins that hydrolyze PPi, releasing the chemical energy of the phosphoanhydride bond as heat, and (b) ion (H^+^ and/or Na^+^)-translocating inorganic pyrophosphatases (mPPases), integral membrane proteins that couple PPi hydrolysis to proton and/or sodium pumping across biological membranes [13,14]. Among the sPPases, two major non-homologous families, known as I and II, have been characterized [15,16,17] (Appendix A). Family I sPPases occur in all types of organisms, both prokaryotic and eukaryotic [18]. In this family of sPPases, PPi binding to the active site is a complex process that depends on divalent cation cofactors, Mg^2+^ being the most efficient. The evolutionarily unrelated family II sPPases, which belong to the DHH-DHHA2 phosphoesterase proteins, are activated by other heavy-metal cations such as Mn^2+^ or Co^2+^ (although Mg^2+^ acts as a co-factor). Family II sPPases include those from *Streptococcus mutans* (a bacterium that contributes to tooth decay in the human oral cavity), *Bacillus subtilis* and other Gram-positive bacteria (Firmicutes), as well as some other lineages of bacteria (Chloroflexi, Spirochaetes, Thermotogae) and archaea (Euryarchaeota). Family I sPPases are strongly inhibited by 0.5 mM fluoride in vitro, whereas much higher concentrations of this anion are needed to inhibit family II sPPases and mPPases [19,20,21].

*Saccharomyces cerevisiae* has two genes encoding sPPases, both belonging to family I: IPP1 and IPP2 (also known as PPA2) (Appendix A). IPP1 is considered an essential nucleocytoplasmic protein while IPP2 is reportedly located in the mitochondrial lumen, where it plays a role in respiratory metabolism [22,23,24]. Previous work from our group in budding yeast showed that cells devoid of IPP1 undergo different fates depending on their energy metabolism; thus, fermenting cultures show massive cell death by autophagy, while respiring cultures do not die but undergo growth arrest in the S-phase of the cell cycle [25]. In human cells, there are two family I sPPase paralogs—PPA1 and PPA2—according to the UNIPROT database. These proteins are located at the cytoplasm and mitochondria, respectively, a situation similar to that of *S. cerevisiae*.

The essentiality of IPP1 for yeast growth, along with previous reports showing the high sensitivity of family I sPPases to fluoride in vitro, prompted us to check whether this interaction could have implications in vivo. The yeast mutant strain YPC3, previously generated in our laboratory [26], was used to accomplish this task. YPC3 cells have their *IPP1* gene (SGD systematic name, YBR011C) under the control of the yeast galactokinase gene (*GAL1*) promoter; consequently, they only express their nucleocytosolic sPPase, when grown on galactose, being unable to grow on other carbon sources. YPC3 cells regain the capacity to grow on glucose (fermentative conditions) or glycerol (respiratory conditions) by transformation with plasmids bearing PPase genes under the control of constitutive promoters [24,25,26,27,28]. In practical terms, YPC3 allows the substitution of IPP1 by different PPi-hydrolyzing enzymes, hence the comparison between different types of PPases within the same cellular context.

Here, we show that the resistance of *S. cerevisiae* cells to fluoride can be increased by overexpressing IPP1 or its human family I ortholog, PPA1. Furthermore, yeast cells in which IPP1 is functionally substituted by relatively low levels of the fluoride-resistant family II sPPase from *S. mutans* may also result in cells with higher tolerance to this anion. These results suggest a major role of IPP1 in the adaptation of *S. cerevisiae* to high concentrations of fluoride and shed light on the cytotoxic effects exerted by this anion at the molecular level. Possible implications for other eukaryotic and prokaryotic cells are discussed.

## 2. Materials and Methods

### 2.1. Bacteria and Yeast Strains

*Escherichia coli* DH5α strain (supE44 lacU169 (Ø80 lacZ M15) hsdR17 recA1 endA1 girA46 thi-1 rel A1) [29] was used for cloning purposes. *S. cerevisiae* haploid strain W303-1A (MATa, ade2-1 can1-100 his3-11,15 leu2-3,112 trp1-1, ura3-1) [30] was used both as a wild type and as a parental strain to generate a YPC3 mutant (W303-1A *ipp1*_UAS_-*ipp1*_TATA_::*HIS3-GAL1*_UAS_-*GAL1*_TATA_-*IPP1*) by the single-step transplacement procedure, as previously described [26].

### 2.2. Plasmids Construction and Yeast Transformation

Plasmids used for yeast transformation, listed in Table 1, were, in most cases, derivatives of the high-copy pRS699b plasmid [26,31] or the low-copy pRS416 plasmid [32]. The coding sequence of the *IPP1* gene was amplified by PCR along with a 400 bp promoter region upstream the start codon, as previously described [26]. The latter has been shown to contain all the cis-acting elements essential for efficient *IPP1* transcription [33]. Artificial *EcoRI* and *SpeI* restriction sites were introduced at the 5′ and 3′ ends, respectively, in order to directionally insert the resulting DNA fragment into the pRS699b plasmid. This yielded plasmid pIPP1-699, in which *IPP1* is inserted between its own promoter and the *PMA1* gene terminator region. The coding sequence of the *S. mutans* family II sPPase (*SPP2*; gene locus DQM59_RS02590) was amplified from genomic DNA by PCR with artificial *EcoRV* and *SpeI* sites at its 5′ and 3′-ends, respectively, and introduced in pIPP1-699 using an *EcoRV* natural restriction site located just upstream the start codon of *IPP1* and the unique *SpeI* site, thus yielding plasmid pSPP2-699. The expression cassettes containing the 400 bp long *IPP1* promoter region, the coding sequences of *IPP1* or *SPP2*, and the *PMA1* terminator region were cleaved out of plasmids pIPP1-699 and pSPP2-699 with restriction enzymes *EcoRI* and *HindIII* and ligated to plasmid pRS416, digested with the same enzymes. This yielded plasmids pIPP1-416 and pSPP2-416. A similar strategy was followed to obtain plasmid piGMVP-416 from a previously published plasmid encoding the ion-translocating membrane PPase (mPPase) of the methanogenic archaeon *Methanosarcina mazei* (MVP) [28] (Appendix A), which is functionally and structurally very different to the sPPases used in this work. High-copy plasmid pPPA1-426, bearing the cDNA coding for the cytosolic sPPase *PPA1* from *Homo sapiens* (PPA1; HGNC, locus NC_000010.11) inserted in the high-copy plasmid pRS426 [34] under the control of the yeast glyceraldehyde-3-phosphate dehydrogenase (*GPD*) promoter, was generously donated by Dr. Agustín Hernández (Betternostic S. L., Noáin, Navarra, Spain). Finally, plasmid pIPPGFP-699 had been previously obtained by our group [24].

*S. cerevisiae* mutant strain YPC3 was transformed with the plasmids described above by using the method of Schiestl and Gietz [35]. Cells were initially grown at 30 °C in rich medium containing galactose [YPGal: 1% (*w*/*v*) yeast extract, 2% (*w*/*v*) peptone, 2% (*w*/*v*) galactose] and transformants were selected by growing cells on 2% (*w*/*v*) agar plates made in synthetic medium [0.17% yeast nitrogen base without amino acids and ammonium sulfate, 2% galactose, 50 mM MES-TRIS pH 6, and a mixture of nucleotides and amino acids, described elsewhere [36], devoid of histidine and uracil].

### 2.3. Phenotype Complementation Tests

For complementation studies, 2 mL of YPD [1% (*w*/*v*) yeast extract, 2% (*w*/*v*) peptone, 2% (*w*/*v*) glucose] was inoculated with transformed cells from the plates and cultured overnight at 30 °C with agitation (200 r.p.m. in a Infors Orbitron orbital shaker) until the stationary phase was reached. An amount of 20 μL of these cultures was used to inoculate 2 mL of YPD and grown again overnight under the same conditions. This treatment was necessary to decrease the pyrophosphatase activity associated with IPP1 in YPC3 cells. For drop tests, ten-fold serial dilutions of the final cultures were made in sterile water, and 2.5 μL drops of each dilution were spotted onto 2% (*w*/*v*) agar plates made in YPD or YPGly (made as YPD with 3% (*w*/*v*) glycerol substituting for glucose). Culture media were buffered by the addition of 50 mM MES (adjusted to the specified pH values with TRIS) and supplemented with NaF at the indicated concentrations. Plates were typically grown at 30 °C for 3–4 days.

### 2.4. Growth Curves in Liquid Medium

Cells were cultured as described above, except that liquid YPD medium adjusted to pH 5.0 with 50 mM MES-TRIS and supplemented with 75 mM NaF was used instead of agar plates. Samples were collected at indicated times and their optical density at 660 nm (OD_660_) was measured.

### 2.5. Preparations of Soluble Protein Extracts from Yeast

Yeast colonies were collected from a plate and liquid-grown up to the stationary phase in selective medium; then, 2–4 mL of YPD was inoculated with a 1:100 volume of stationary culture. After overnight growth at 30 °C with agitation as described above, cells were sedimented by centrifugation in a bench centrifuge (3000× *g*, 5 min), washed thoroughly with water, resuspended in 0.2 mL of ice-cold buffer A (25 mM Tris-HCl, pH 8, 10% (*w*/*v*) glycerol, 2 mM DTT, 1 mM EDTA, 1 mM benzamidine, 2 mM ε-aminocaproic acid, 1 mM PMSF), and homogenized by vigorous shaking with glass beads. The homogenate was diluted up to 0.5 mL with buffer B (10 mM Tris-HCl, pH 8, 10% glycerol, 2 mM DTT, 1 mM EDTA) and centrifuged in a microfuge (20,000× *g*, 20 min) at 4 °C to remove beads, debris, and most of the membrane fraction. The resulting supernatant was used as the soluble protein extract.

### 2.6. Enzymatic Assay, Protein Determinations and Western Blotting

Pyrophosphatase activity was assayed at 30 °C in a final volume of 200 mL in a medium containing 50 mM MOPS-TRIS pH 7.2, 1 mM MgCl_2_ and 0.5 mM Na_4_PPi in the presence and absence of 0.5 mM NaF. The amount of released orthophosphate was determined spectrophotometrically, as previously described [37]. In every assay, the amount of sample and/or the incubation time (usually 3–5 min) were adjusted so that the absorbance values fell within the linear part of a calibration curve made with a commercial orthophosphate standard solution. Absorbances were measured every 20 s, plotted versus time, and the activities were calculated from the slopes of the resulting straight lines. Protein concentration was estimated using a Coomassie-blue dye binding-based assay from Bio-Rad (München, Germany) according to the manufacturer’s instructions using ovalbumin as a standard. Immunodetection by Western blot was performed as described elsewhere [38] using a commercial rabbit polyclonal antibody against *S. cerevisiae* IPP1 (AP21326F_N, OriGene EU, Herford, Germany).

### 2.7. Determination of Internal PPi Levels in Yeast Cells

YPC3 cells transformed with the plasmids shown in Table 1 were grown in YPD buffered at pH 5, as described above. The initial OD_660_ values of the cultures were individually adjusted so that after 8–9 h, all of them had a final OD_660_ of around 0.5 (roughly 10^9^ cells). Cultures were then split into two aliquots, and NaF (final concentration 75 mM) was added to one of them. Cells were grown for two more hours under the same conditions, after which they were collected by centrifugation, washed with ice-cold deionized water and broken as described in Section 2.5, except that a 4% (*v*/*v*) perchloric acid aqueous solution was used instead of buffer A. Beads, debris, and denatured proteins were removed by centrifugation (20 min, 20,000× *g*, 4 °C). PPi concentrations were determined in the supernatants as previously described [39].

**Table 1 microorganisms-13-00226-t001:** Plasmids used in this communication.

Plasmid	Description	References
pRS699b ^1^	*S.cerevisiae/E. coli* shuttle plasmid bearing the constitutive promoter and terminator of yeast gene *PMA1* and selection marker *URA3*. It has a 2-micron (2 m) origin of replication, which yields a high number of plasmid copies (14–34) per cell.	[31]
pIPP1-699 ^1,2^	Plasmid derived from pRS699b bearing the promoter and coding sequence of gene *IPP1* from *S. cerevisiae* inserted upstream of the *PMA1* terminator.	[40]
pIPP1-416 ^1^	Plasmid derived from pRS416 bearing the promoter and coding sequence of gene *IPP1* from *S. cerevisiae*. Its main difference from pIPP1-699 is that it has a yeast centromeric sequence (CEN) and an autonomously replicating sequence (ARS) that results in a low number of plasmid copies (2–5) per yeast cell.	This study
pSPP2-699 ^1^	Plasmid derived from pIPP1-699 bearing gene *SPP2* from *S. mutans*.	This study
pSPP2-416 ^1^	Plasmid derived from pIPP1-416 bearing gene *SPP2* from *S. mutans*.	This study
pPPA1-426 ^1^	Plasmid derived from high-copy plasmid pRS426 that bears the cDNA coding for the cytosolic sPPase from *Homo sapiens* (PPA1).	Donated by Dr. Hernández
piGMVP-416 ^1,2^	Plasmid derived from pIPP1-416 bearing the sequence that codes for the putative N-terminal signal peptide of yeast Suc2p followed by those of yEGFP and the Na^+^-translocating mPPase from *M. mazei* (MVP).	This study and [28]
pIPP1GFP-699 ^2^	Plasmid derived from pIPP1-699 bearing the coding sequence of gene *IPP1* from *S. cerevisiae* followed in-frame by that of yEGFP.	[24]

^1^ Plasmid used for complementation and fluoride resistance studies. ^2^ Plasmid used for transformation method based on selection in YPD.

### 2.8. Spontaneous Generation of Fluoride-Resistant Cells

W303-1A cells were grown up to the exponential phase (OD_660_ between 0.6 and 0.9) in liquid unbuffered YPD medium devoid of NaF; then, ten-fold serial dilutions of these cultures were carried out in sterile water and 100 μL of each dilution was extended onto 2% (*w*/*v*) agar plates made in YPD buffered at pH 5 and supplemented with 60 to 70 mM NaF. Plates were incubated at 30 °C for at least one week.

### 2.9. Transformation of YPC3 Cells and Selection of Transformants in YPD

Transformation was performed by the method of Schiestl and Gietz [35] with some modifications: 2 mL of YPGal was inoculated with YPC3 cells from an agar plate prepared in the same medium and cultured overnight at 30 °C with agitation, as in Section 2.3. The following day, 15 mL of YPGal was inoculated with the overnight stationary culture so that the initial OD_660_ was around 0.3. Cells were grown for 4 h under the same conditions, collected by centrifugation, resuspended in 15 mL of YPD, and grown for 2 more hours until the OD_660_ was between 1 and 1.3. The last step was necessary to allow YPC3 cells to adapt from galactose to glucose metabolism before the transformation/selection procedure.

Glucose-adapted YPC3 cells were collected by centrifugation (3000× *g*, 1 min in a bench centrifuge), washed with 15 mL of sterile water, resuspended in 1 mL of sterile water, transferred to a microtube, and centrifuged again in a microfuge (13,000× *g*, 30 s). The final washed pellet was resuspended in 1 mL of sterile water, and 100 mL aliquots were transformed with plasmids pRS699b (control), pIPP1-699, pIPP1GFP-699 and piGMVP-416 by the lithium acetate/polyethylene glycol method [35]. After heat-shock lasting 20–25 min at 42 °C, transformed cells were sedimented in a microfuge and resuspended in 1 mL of sterile water, and 20 μL aliquots were extended onto YPD agar plates buffered with 50 mM MES-TRIS, pH 5, with the optional addition of NaF (up to 40 mM final concentration).

### 2.10. Fluorescence Microscopy

Individual colonies of YPC3 transformed as described in the previous paragraph were grown in liquid YPD for 4 h and directly visualized with a fully automated Leica DM6000B microscope (Leica Microsystems) with FITC green fluorescence filters (excitation filter 480/40 nm, dichromatic mirror 505 nm, suppression filter 527/30 nm), a 40× objective, and equipped with a cooled CCD (charge-coupled device) camera (ORCA-AG, Hamamatsu Photonics).

## 3. Results

YPC3 cells were transformed with plasmids bearing gene coding for different evolutionarily unrelated PPases under the control of constitutive promoters (Table 1).

Transformants recovered the capacity to grow under fermentative conditions (2% (*w*/*v*) glucose) with the only exception being cells transformed with plasmid pRS699b. Moreover, YPC3 cells transformed with plasmids pIPP1-699, pSPP2-699, pSPP2-416 and piGMVP-416 could also grow in the presence of up to 75 mM NaF, unlike the parental strain W303-1A. Interestingly, cells transformed with the plasmids bearing the gene coding for the fluoride-insensitive family II sPPase from *S. mutans* showed the highest tolerance to this anion. Plasmids pIPP1-416 and pPPA1-426 were less effective at conferring resistance to this salt (Figure 1). Identical results were obtained with KF, demonstrating that the toxicity of NaF was a specific effect of fluoride.

The results obtained with YPD agar plates were confirmed by growing cells in liquid culture media (Figure 2). For the sake of clarity, only the growth of W303-1A cells and strain YPC3 transformed with plasmid pIPP1-699 in liquid YPD, pH 5.0, with 75 mM NaF is shown, but similar results were obtained with YPC3 cells transformed with plasmids pSPP2-699 and pSPP2-416.

YPC3 cells transformed with plasmids bearing different PPases also increased fluoride resistance in agar plates with medium containing glycerol, considered a ‘non-fermentable’ carbon source in yeast [41] (Figure 3).

Crude extracts from YPC3 cells transformed with plasmid pIPP1-699 showed a 10-fold increase in soluble PPase-specific activity as compared to those obtained from the parental strain W303-1A, the increase being only around 50% higher in the case of cells transformed with centromeric plasmid pIPP1-416. YPC3 cells transformed with high-copy plasmid pPPA1-426, bearing the gene coding for the human cytosolic sPPase, also showed a significant increase in soluble PPase activity (around three-fold) with respect to W303-1A cells. In all these cases, the respective activities were strongly inhibited by the presence of 0.5 mM NaF in the assay (Figure 4).

Lower levels of specific PPase activity, between 5- and 10-fold less than those observed with multi-copy plasmids bearing family I sPPases, were detected in extracts obtained from cells transformed with pSPP2-699 and pSPP2-416, which encode a family II sPPase. In these cases, the soluble PPase activity was insensitive to fluoride (Figure 4).

Cells transformed with piGMVP-416, encoding the Na^+^-translocating mPPase from *M. mazei* (MVP), only showed membrane-associated fluoride-insensitive PPase activity that increased by six- to seven-fold in the presence of 100 mM KCl, as previously described [28].

Immunodetection performed in cell extracts with a polyclonal antibody against *S. cerevisiae* IPP1 showed that, in YPC3 cells transformed with plasmids pIPP1-699, pIPP1-416 and pPPA1-426, increases in PPase activity correlated with the levels of a polypeptide of 32 kDa, the expected size for eukaryotic family I sPPase polypeptides [18]. No band was detected in cells transformed with control plasmid pRS699b or with pSPP2-699, pSPP2-416, and piGMVP-416 (Figure 5).

Internal levels of PPi significantly increased in soluble extracts of W303-1A cells and YPC3 cells transformed with plasmid pIPP1-416 after growing for only 2 h in the presence of NaF. The overexpression of family I PPases or expression of the *S. mutans* family II PPase SPPA2 prevented this scenario (Figure 6).

Fluoride-resistant yeast cells spontaneously appeared after the growth of W303-1A cells for 6–7 days in YPD agar plates buffered at pH 5 and supplemented with 70 mM NaF. These experiments were carried out as described in the Section 2 with six independent W303-1A clones. Around 0.003% of cells were estimated to spontaneously develop fluoride resistance, none of them showing higher specific sPPase activity than the parental strain.

YPC3 cells were transformed with plasmids pIPP1-699 and pIPP1GFP699 and extended onto YPD plates. Yeast colonies were observed in the plates after 36 h, whereas none appeared in the case of cells transformed with plasmid pRS699b (Figure 7, upper panel). Three independent colonies were randomly selected from each plate and all of them were able to grow on YPD agar plates supplemented with 60–75 mM NaF and buffered at pH 5. Alternatively, up to 20 mM NaF (along with 50 mM MES-TRIS pH 5) were directly added to the YPD agar plates used for selection, although this resulted in a reduced number of transformants (Figure 7 lower panel). The same procedure was used with plasmid piGMVP-416, but in this case, transformants appeared after 60 h.

YPC3 cells transformed with plasmids pIPP1GFP-699 and piGMVP-416 and selected in YPD were visualized by fluorescence microscopy, showing nucleocytosolic or membrane-associated green fluorescence, respectively, unlike cells transformed with pIPP1-699 (Figure 8).

## 4. Discussion

The aim of this project was to establish a possible role of IPP1, the essential nucleocytosolic sPPase of *S. cerevisiae*, in the adaptation of this organism to high concentrations of fluoride. This idea was based on previous evidence demonstrating that, on the one hand, this anion is toxic for yeast [5], and, on the other hand, that it is a strong inhibitor of IPP1 in vitro [42]. Initially, *S. cerevisiae* haploid strain W303-1A was transformed with the plasmids described in Table 1, and the resulting transformants were able to grow on plates with higher concentrations of NaF than cells transformed with a control. Previous reports used a similar approach, but using yeast strain BY4741 as a wild-type, obtaining opposite results [43]. In both cases, transformants obtained from W303-1A or BY4741 cells simultaneously expressed two sets of PPases, one encoded by the native gene located at yeast chromosome 2 (https://frontend.qa.yeastgenome.org/locus/S000000215, accessed on 20 December 2024), and the other by the plasmid-borne copies. This hindered the capacity to control nucleocytosolic PPase activity and precluded the possibility to test other PPases (see below for further discussion).

The situation described above can be circumvented by using the yeast mutant YPC3, whose chromosomal gene *IPP1* is under the control of the galactose-inducible glucose-repressible *S. cerevisiae* galactokinase (*GAL1*) promoter [26]. When YPC3 cells are grown on glucose or glycerol transcription of *IPP1* is stopped, levels of nucleocytosolic PPase hydrolytic activity become negligible after several hours, and cellular growth stops [24,25]. YPC3 cells transformed with plasmids bearing genes coding for different types of PPases (Table 1) not only recovered the capacity to grow on glucose, as previously described [24,26,28], but in most cases they could also grow in the presence of fluoride concentrations significantly higher than the parental strain W303-1A (Figure 1). Sensitivity to fluoride was higher at lower pH values in all cases, thus supporting the idea that this anion enters the cells as HF (pKa 3.14), as previously reported [44]. The fact that cells transformed with control plasmid pRS699b could not grow on YPD further indicates that the PPases encoded by the plasmid-borne genes are mandatory to support growth in this glucose-containing medium.

The levels of fluoride-sensitive sPPase activity detected in crude soluble extracts obtained from cells transformed with plasmids bearing *IPP1* correlated with their capacity to grow in the presence of high concentrations of NaF. The high levels of PPase activity measured in cells transformed with pIPP1-699 were consistent with this plasmid bearing the two-micron (2μ) origin of replication, residing in the yeast nucleus at an estimated copy number of 14 to 34 per haploid cell. By contrast, plasmid pIPP1-416 gives only two to five copies of *IPP1* per cell because it contains an autonomously replicating sequence (ARS) and a yeast centromere sequence (CEN) instead of the 2μ origin of replication [32,45].

Transformation with the 2 μ containing plasmid pPPA1-426 increased sPPase levels with respect to those of W303-1A but not by as much as those of pIPP1-699. There may be several reasons for this: (a) the *GPD* promoter in this plasmid might produce lower transcription levels than the *IPP1* promoter present in the rest of the plasmids; (b) the translation of IPP1 mRNA is more effective than that of PPA1; and/or (c) intracellular conditions are more adequate for IPP1 than for the heterologous PPA1. In any case, both IPP1 and PPA1 activities were strongly inhibited by 0.5 mM fluoride in vitro, as expected from molecular phylogenetic analyses that showed that the genes coding for these proteins are evolutionary not distant orthologs, both belonging to the eukaryotic assembly of family I sPPases (Appendix A). Structural studies have further established the high similarity between IPP1 and PPA1 [46].

The results obtained with SPP2 and MVP demonstrate that very high levels of fluoride-sensitive PPase activity are not mandatory to gain tolerance to this anion; indeed, low (or even very low) levels of fluoride-insensitive activity are enough. In other words, yeast cells require a minimal threshold of PPase activity in any situation regardless of the enzymatic system that accomplishes this task. These results further suggest that yeast strain W303-1A has significantly higher levels of cytosolic PPase activity than those necessary for normal growth on YPD.

Growth tests on YPGly demonstrate that IPP1, rather than IPP2 (the yeast mitochondrial PPase), is the limiting factor for growth in the presence of this anion as well as under respiratory conditions. IPP2 has been shown to be as sensitive to fluoride as IPP1 in vitro; therefore, our results probably reflect the fact that HF, the actual molecule that permeates membranes, is mostly dissociated into F^-^ ad H^+^ in yeast cytosol, which is normally maintained at pH values of 7.2–7.4 [47]. In respiring cells, IPP1, SPP2, and MVP were equally effective at supporting growth at pH 5 and 75 mM NaF regardless of the type of plasmid. A possible explanation for this effect is that the rate of IPP1 depletion in YPC3 differs between fermentative and respiratory conditions; in fact, it takes 24 h to stop detecting IPP1 after the cells are transferred from galactose to glycerol, whereas only 6 h is needed when switching from galactose to glucose [25]. This timing difference may represent a crucial period duration in which to adapt to the presence of fluoride (and presumably to other stress conditions).

Cells transformed with plasmid pPPA1-426 had significantly worse growth both in the absence and presence of NaF, which suggests that respiratory conditions might exert negative effects on PPA1 activity in vivo via post-translational modifications or changes in the intracellular milieu, among other possible reasons. It should be noted in this respect that the ubiquitination of IPP1 has been linked to the regulation of its levels in different subcellular compartments [24].

Immunodetection with a polyclonal antibody against IPP1 indicated a positive correlation between fluoride-sensitive PPase activity and levels of enzyme polypeptides in cells expressing family I sPPases. As expected, the antibody could not recognize the sPPase II from *S. mutans*, a protein structurally very different from IPP1 [21]. This was also the case for MVP, as previously described [28].

Measurements of intracellular PPi concentrations predictably showed that there is an inverse relationship between intracellular levels of this metabolite and PPase activity levels. YPC3 cells transformed with pRS699b that stopped growing after six hours in YPD always showed very high levels of intracellular PPi, a fact that has been previously reported [28]. Similar levels were observed in W303-1A cells and YPC3 cells transformed with pIPP1-416 after two hours in the presence of 75 mM NaF. These results strongly suggest that in these cells, fluoride inhibits IPP1 up to the point at which PPi accumulates, stopping cell growth by collapsing anabolism [12,25]. Internal PPi concentrations for YPC3 expressing MVP can be found elsewhere [28].

The results presented here show that resistance to fluoride in *S. cerevisiae* can be achieved either by increasing the levels of a fluoride-sensitive PPase activity or by expressing an intrinsically fluoride-resistant PPase in the cytosol (or indeed embedded into membranes). This further suggests that IPP1 is an important target of fluoride in vivo in budding yeast, both under fermentative and respiratory conditions.

In a previous report, the overexpression of IPP1 only conferred fluoride resistance to a fluoride exporter (FEX) double-knock-out mutant but not to its parental strain; however, there were significant differences between those experiments and the ones reported here: (1) yeast strains have different genetic backgrounds (W303-1A vs. BY4741); (2) we used a yeast mutant in which the levels of IPP1 could be brought down to negligible levels by changing the carbon source in the culture medium, thus allowing us to qualitatively and quantitatively control PPase activity in the (nucleo)cytosol by using appropriate plasmids; (3) we used YPD buffered with 50 mM MES to control external pH; and (4) *IPP1* was under the control of its own promoter in our plasmids, whereas the *GPD* promoter was used in the aforementioned report [43].

The fact that the metabolic scenario in human cells concerning sPPases is comparable to that in *S. cerevisiae* prompts us to speculate that fluoride might also act by inhibiting the cytosolic sPPase in the former, thus increasing PPi concentrations in this cell compartment. However, unlike yeast, some types of human cells (chondrocytes, cartilage cells) present an integral membrane protein (ANK) that reportedly exports PPi outside the cell, thus contributing to maintain substantial extracellular levels of this metabolite [48]. An abnormal accumulation of intracellular PPi produced by the inhibition of PPA1 might result in a higher rate of extrusion from the cell via ANK, thereby increasing extracellular PPi concentration and directly affecting hydroxyapatite formation in bones, teeth and other tissues. Moreover, PPi has been reported to act as a signaling molecule that may also alter biomineralization by influencing gene expression and regulating its own production and breakdown [48,49]. Therefore, the results presented in this communication suggest a molecular mechanism by which fluoride might be involved in the pathogenesis of some human diseases such as diseases of the teeth and skeletal fluorosis. On the other hand, certain tumor cells have been reported to enrich their surrounding environment with lactic acid generated by fermentation (the Warburg effect), while upregulating the expression of PPA1 [50,51]. Based on the results presented here, we can hypothesize that fluoride might diffuse within these cells quite effectively due to the acidic environment, thereby inhibiting their sPPase activity in vivo and increasing internal PPi levels. This situation would presumably alter their anabolism, cell cycle, and proliferation [25,52]

Our results also suggest that prokaryotes (bacteria and archaea) such as *S. mutans* might be more resistant to fluoride than others that have fluoride-sensitive family I sPPases (for example, *E. coli*) [18]; therefore, the use of fluorinated toothpaste or water could alter our oral and intestinal microbiota in favor of microorganisms that have genes coding for family II sPPases in their genomes.

Attempts to develop a method for selecting transformants of wild-type yeasts in YPD based on the increase in fluoride resistance induced by the overexpression of IPP1 were unsuccessful. The transformation of W303-1A cells with plasmids bearing PPases followed by selection in pH 5-buffered YPD supplemented with 60 to 70 mM NaF resulted in a small number of colonies appearing after several days in all cases, including for the control plasmid pRS699b. This demonstrated that it was possible for fluoride resistance to spontaneously arise in our strains and led us to study this effect more in detail using our parental strain (Section 2.9). The fact that none of the spontaneous mutants obtained had more PPase activity than the original cells supports previous reports showing that yeast cells have alternative ways to develop resistance to fluoride [8,43] and raises the question of what role IPP1 actually plays in this process.

PPi hydrolysis has been suggested to be “an ancient mechanism that imparts irreversibility (...) functioning like a ratchet’s pawl to vectorialize the life process toward growth” [53]. This means that increasing levels of fluoride in the cytosol would gradually inhibit multiple enzymatic systems, including IPP1 itself, which in turn would slow down anabolism, hindering the capacity of the cell to respond. Maintaining adequate PPi levels in the cytosol might prevent, or at least delay, anabolic collapse, thus giving the cell crucial time to activate the different mechanisms that help cope with fluoride toxicity [8].

Finally, in this communication, we describe a method for a cheap and rapid selection of yeast transformants based on mutant YPC3 and the use of PPases as selection marker genes. YPC3 cells can be transformed with plasmids bearing different types of PPases and the resulting transformants selected in YPD. The latter can be further checked by their capacity to grow in buffered medium at pH 4.5–5 supplemented with NaF, the level and/or characteristics of their PPase activities, and/or by fluorescence microscopy (if chimeric constructs of GFP-fusioned PPases are used in the plasmids). Transformants can also be directly selected in pH 5-buffered YPD supplemented with up to 20 mM NaF with a gradual reduction in the number of transformants obtained. The combination of glucose-containing medium and resistance to NaF can be used as a form of selection pressure to maintain the plasmids in transformed YPC3 cells. This selection method is rapid and uses an easy-to-prepare standard medium that can be supplemented with a cheap chemically stable biocide. The optional use of NaF in YPD plates can also prevent contamination by organisms like *E. coli*, often present in laboratory environments, which makes this method suitable for yeast work in places with limited resources to control culture media sterility.

## 5. Conclusions

The sensitivity to fluoride of the nucleocytosolic PPase from yeast (IPP1) has implications in vivo.Fluoride resistance in yeast requires a minimal threshold of PPase activity. This can be accomplished with different types of PPases.The results presented here may have implications for other systems such as human cells and bacteria involved in tooth decay.A cheap, rapid, and easy method for the selection of YPC3 transformants in YPD optionally supplemented with fluoride is described.

## Figures and Tables

**Figure 1 microorganisms-13-00226-f001:**
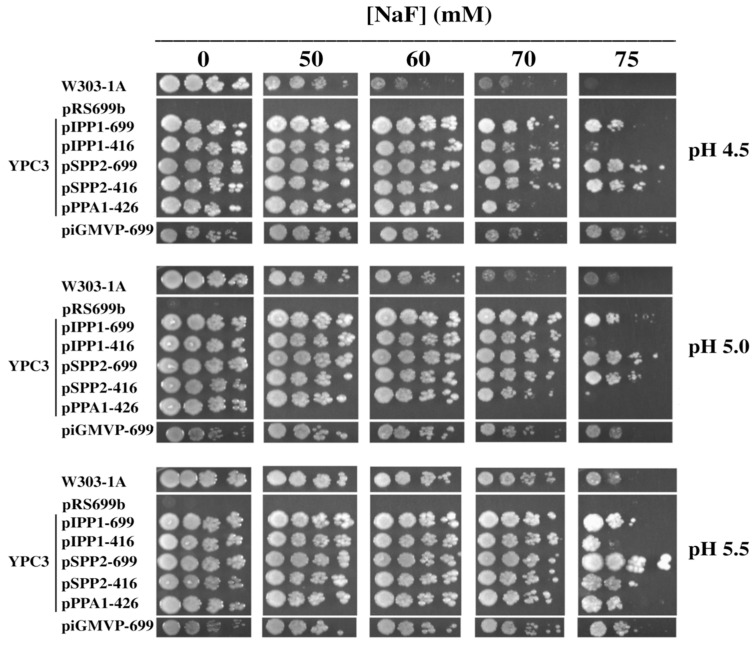
Drop tests of transformed YPC3 cells and parental strain W303-1A grown at different pH values and concentrations of NaF under fermentative conditions. YPC3 cells were transformed with plasmids shown in Table 1 and grown as described in the main text. Serial dilutions of the cultures were prepared in sterile water, spotted onto YPD agar plates containing 50 mM MES (adjusted to the specified pH values with TRIS) and supplemented with the indicated concentrations of NaF. Growth was recorded after 3 days, except for plates containing 75 mM NaF, which were grown for 4 days before the photograph was taken. Cell viability declined with NaF concentrations above 75 mM until the complete halting of growth was observed at 100 mM NaF.

**Figure 2 microorganisms-13-00226-f002:**
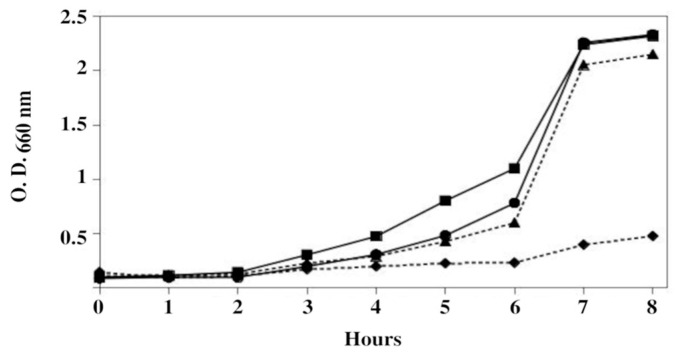
Growth of W303-1A cells and YPC3 cells transformed with plasmid pIPP1-699 in liquid YPD adjusted to pH 5.0 in the absence and presence of 75 mM NaF. (-■-) W303-1A cells grown without NaF, (-●-) YPC3 cells transformed with plasmid pIPP1-699 without NaF, (--♦--) W303-1A grown with 75 mM NaF, and (- -▲- -) YPC3 cells transformed with plasmid pIPP1-699 grown with 75 mM NaF. The OD_660_ values are the averages of three independent experiments.

**Figure 3 microorganisms-13-00226-f003:**
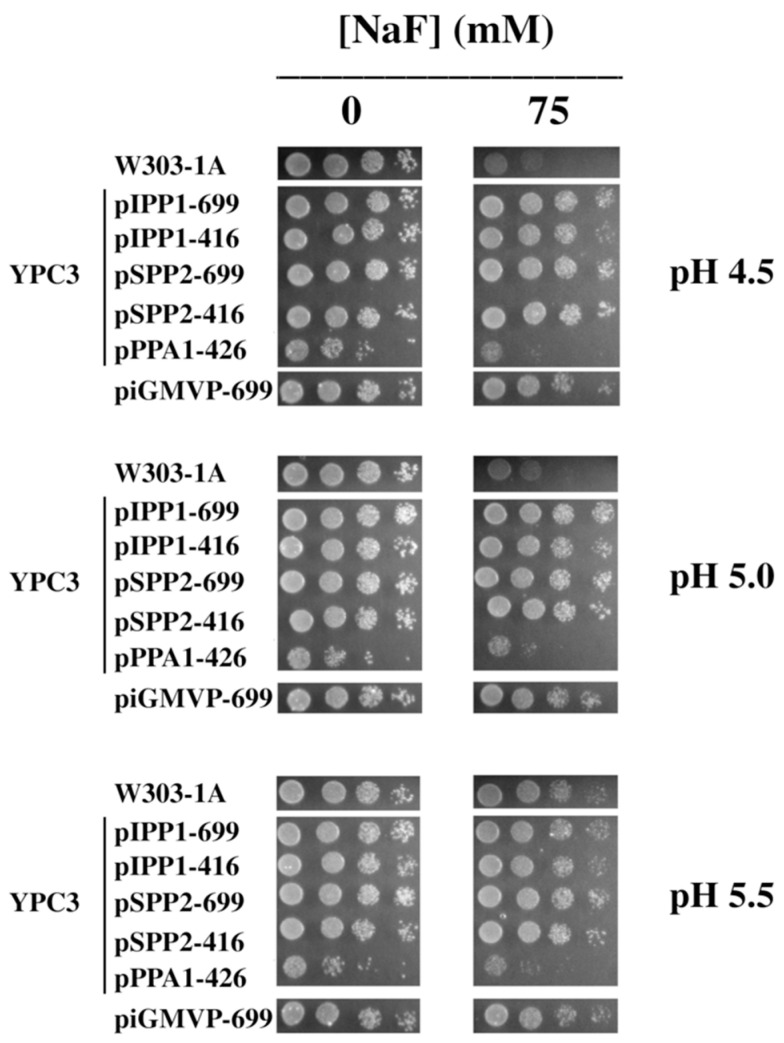
Drop tests of transformed yeast mutant YPC3 cells and parental strain W303-1A grown at different pH values and concentrations of NaF under respiratory conditions. Experiments were performed as in Figure 1, except that 3% (*w*/*v*) glycerol substituted for glucose in the plates.

**Figure 4 microorganisms-13-00226-f004:**
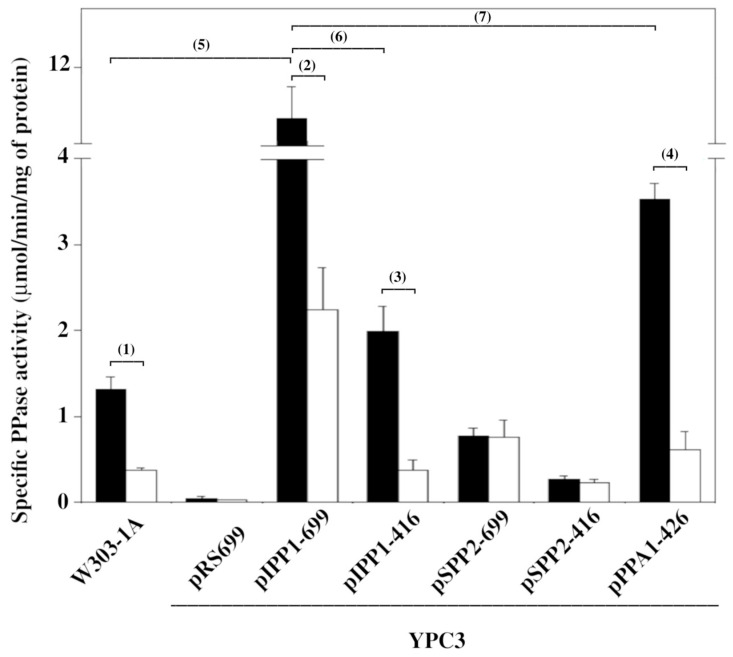
Levels of hydrolytic sPPase activity in protein extracts observed from transformed yeast mutant YPC3 cells and parental strain W303-1A. Activity assays were performed in the absence (black columns) and presence (white columns) of 0.5 mM NaF, as described in the Section 2. Values of specific activities are averages ± SE corresponding to at least 5 independent experiments. Student’s unpaired *t* tests were performed using the *T* test calculator on the webpage https://www.graphpad.com/quickcalcs/ttest1/?format=SEM (accessed on 10 January 2025). Legend: (1) statistically significant difference (*p* = 0.0133), (2) very statistically significant difference (*p* = 0.0042), (3) very statistically significant difference (*p* = 0.0024), (4) extremely statistically significant difference (*p* = 0.0008), (5) very statistically significant difference (*p* = 0.0045), (6) very statistically significant difference (*p* = 0.0070), and (7) statistically significant difference (*p* = 0.0211).

**Figure 5 microorganisms-13-00226-f005:**
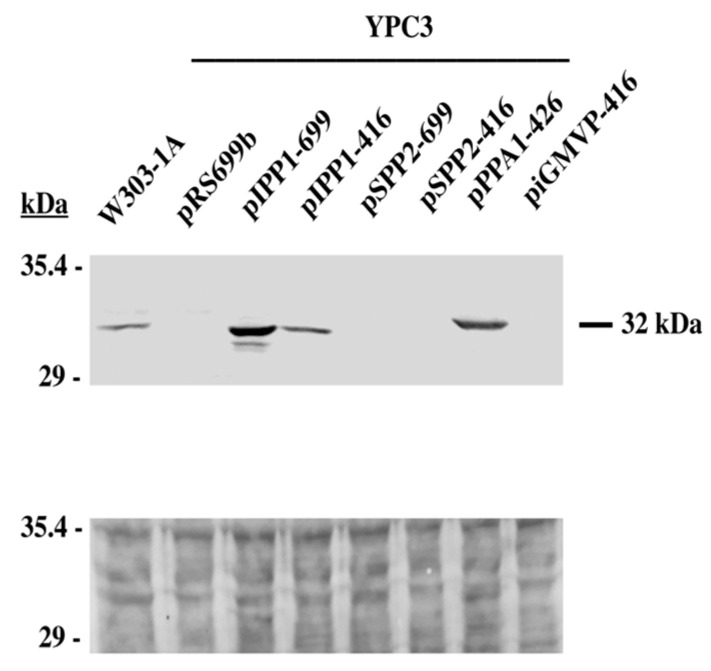
Immunodetection of family I sPPases in protein extracts obtained transformed YPC3 cells and parental strain W303-1A (upper panel). A polyclonal antibody against *S. cerevisiae* IPP1 was used. In total, 50 μg of total protein was loaded per lane. The lower panel shows the Ponceau S staining of the nitrocellulose filter after transferring the proteins from the SDS-PAGE gel.

**Figure 6 microorganisms-13-00226-f006:**
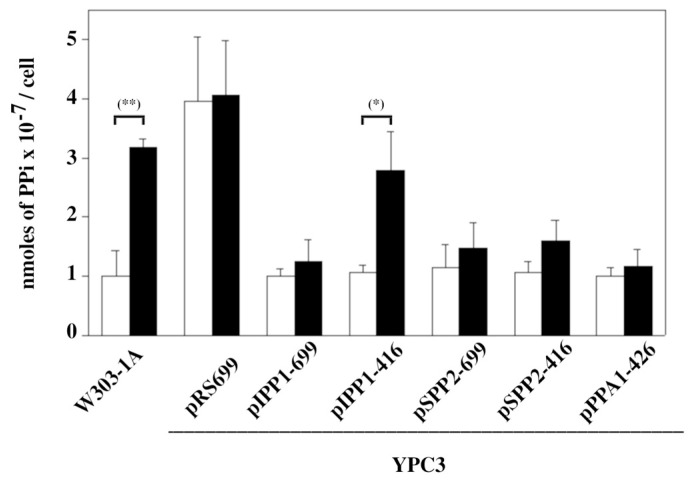
Internal levels of PPi measured in transformed YPC3 cells and parental strain W303-1A after 2 h of growing in the presence of 75 mM NaF (black columns). White columns show PPi levels of the respective control cells grown in standard YPD (see the Section 2 for further details of the procedure). Values are averages ± SE corresponding to 4 independent experiments. Student’s unpaired *t* tests were performed using the *T* test calculator at on webpage https://www.graphpad.com/quickcalcs/ttest1/?format=SEM (accessed on 10 January 2025) Legend: (*) statistically significant difference (*p* = 0.0457) and (**) very statistically significant difference (*p* = 0.0025).

**Figure 7 microorganisms-13-00226-f007:**
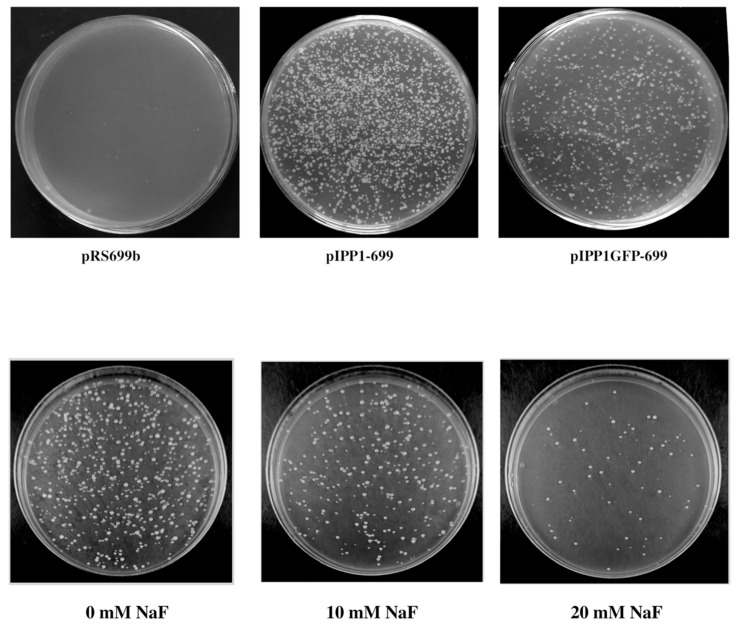
Transformed YPC3 cells selected in YPD agar plates (**upper panel**) and in buffered YPD, pH 5, supplemented with the indicated concentrations of NaF (**lower panel**). Cells were transformed as described in the Section 2 using plasmids pRS699b (negative control), pIPP1-699 and pIPP1GFP-699 (**upper panel**), and pIPP1-699 (**lower panel**). Photographs were taken 36 h after transformation.

**Figure 8 microorganisms-13-00226-f008:**
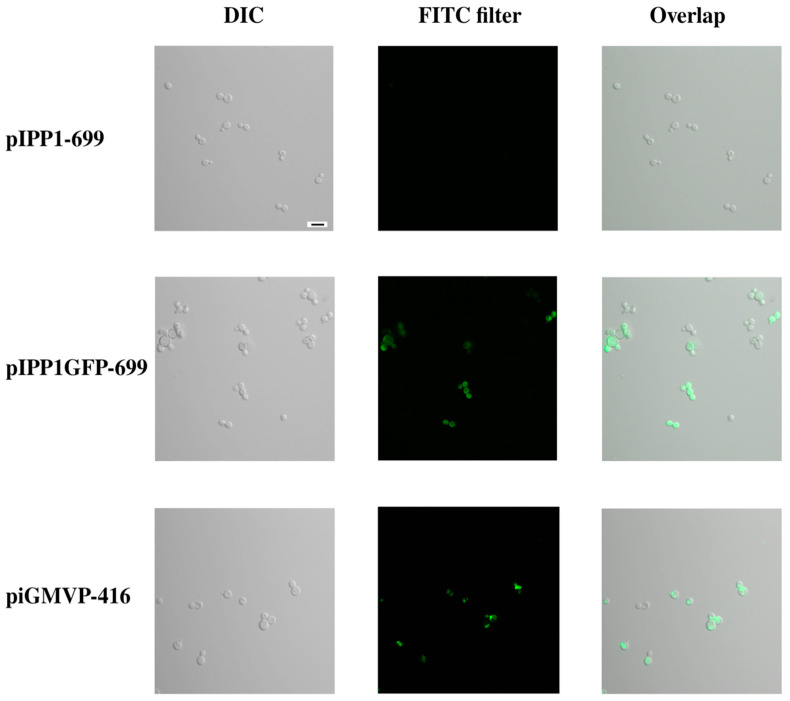
In situ fluorescence microscopy of YPC3 cells transformed with plasmids pIPP1-699, pIPP1GFP-699, and piGMVP-416. Cells were visualized in a Leica DM6000B microscope. The left-hand column shows the DIC microscopy of a typical field of transformed cells, the middle column shows the pattern of intracellular fluorescence distribution (observed with a FITC green filter) within the same cells, and the right-hand column shows the overlap of both. The black bar in the top-left photograph indicates 10 mm.

## Data Availability

The original contributions presented in this study are included in the article/Appendix A. Further inquiries can be directed to the corresponding author.

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
