# Peer review of "Engineering Inorganic Pyrophosphate Metabolism as a Strategy to Generate a Fluoride-Resistant Saccharomyces cerevisiae Strain"

_microorganisms, 2025, doi:10.3390/microorganisms13020226_

Round 1
Reviewer 1 Report
Comments and Suggestions for Authors
The results presented in the manuscript demonstrate that inorganic pyrophosphatase (PPase) inhibition in yeast cells is the main cause of fluoride toxicity and also show the ways to overcome this toxicity by boosting PPase synthesis or replacing yeast PPase with its more tolerant versions. The findings of this study may have a biotechnological impact. I have only minor remarks on this otherwise elegant, well-planned, and carefully done study.
1. The authors should provide a more detailed description of the procedures used, in particular those involving long incubations. The durations of these incubations may be important for their reproduction by readers and should be indicated. The reason is that some reactions involving fluoride are not instantaneous but are relatively slow. They include PPase inactivation by fluoride and reactivation after lowering fluoride concentration (by dilution, for example). As a result, PPase activity measured in the presence of fluoride using a fixed-time assay depends on the assay time because the product Pi will accumulate non-linearly. If PPase is inhibited in the cells and they are then broken and diluted, PPase will gradually reactivate (with t1/2 of approx. 1 h). Furthermore, at the concentrations of fluoride and Mg2+ used, MgF2 solubility product (10^(-8.2) Kohlrausch, F. (1908) Z. Phys. Chem. 64, 129-169) is exceeded and MgF2 precipitation is expected, but may not occur substantially as it is a slow process.
2. Line 67 – replace “equilibria” with “reactions” because thermodynamic equilibrium cannot be shifted.
3. Line 80 – or.
4. Line 81 – replace “co-substrate” with “cofactor”.
5. Line 216 and elsewhere – replace rpm with g value (rpm cannot be reproduced on a different centrifuge).
6. Line 227 – delete “with a”.
7. Figure 6 - define the meaning of blank and filled bars in the figure legend.
Author Response
The results presented in the manuscript demonstrate that inorganic pyrophosphatase (PPase) inhibition in yeast cells is the main cause of fluoride toxicity and also show the ways to overcome this toxicity by boosting PPase synthesis or replacing yeast PPase with its more tolerant versions. The findings of this study may have a biotechnological impact. I have only minor remarks on this otherwise elegant, well-planned, and carefully done study.
- The authors should provide a more detailed description of the procedures used, in particular those involving long incubations. The durations of these incubations may be important for their reproduction by readers and should be indicated. The reason is that some reactions involving fluoride are not instantaneous but are relatively slow. They include PPase inactivation by fluoride and reactivation after lowering fluoride concentration (by dilution, for example). As a result, PPase activity measured in the presence of fluoride using a fixed-time assay depends on the assay time because the product Pi will accumulate non-linearly. If PPase is inhibited in the cells and they are then broken and diluted, PPase will gradually reactivate (with t1/2 of approx. 1 h). Furthermore, at the concentrations of fluoride and Mg2+ used, MgF2 solubility product (10^(-8.2) Kohlrausch, F. (1908) Z. Phys. Chem. 64, 129-169) is exceeded and MgF2 precipitation is expected, but may not occur substantially as it is a slow process.
First of all, we want to thank referee 1 his comments, that will undoubtedly help to improve our manuscript.
Comment #3 is a very good point. We have described more in detail the procedure we follow to assay PPase activity in Materials and Methods. We hope this meets the requirements by referee 1
- Line 67 – replace “equilibria” with “reactions” because thermodynamic equilibrium cannot be shifted.
Done
- Line 80 – or.
Corrected
- Line 81 – replace “co-substrate” with “cofactor”.
Done
- Line 216 and elsewhere – replace rpm with g value (rpm cannot be reproduced on a different centrifuge).
Changed when referred to centrifugation, maintained when referred to agitation in orbital shaker (we have written the brand and model of the shaker)
- Line 227 – delete “with a”.
Done
- Figure 6 - define the meaning of blank and filled bars in the figure legend.
Done
Reviewer 2 Report
Comments and Suggestions for Authors
General Comments
The manuscript explores a novel approach to improving fluoride resistance in yeast through genetic manipulation of inorganic pyrophosphatases. The study addresses an important issue, given fluoride’s widespread presence in the environment and its toxic effects on living organisms, including microbes. The authors successfully employ genetic engineering to enhance fluoride resistance using a combination of overexpressed fluoride-sensitive and fluoride-insensitive pyrophosphatases, and they propose an innovative selection method based on this resistance. While the topic is timely and relevant to both environmental science and synthetic biology, several significant issues reduce the impact and reliability of the manuscript. The experimental design lacks critical controls, and the statistical rigor required to validate findings is insufficient. The absence of gene expression quantification and biochemical evidence to support proposed mechanisms limits the scientific depth of the study. Furthermore, the ecological and industrial relevance of the findings is underexplored, and the data presentation is inconsistent and difficult to interpret. Thus, the manuscript requires major revisions to address the issues outlined below. The authors should prioritize experimental validation of key findings, provide a clearer link between their results and the broader field, and improve the clarity and organization of the text.
Major Concerns
#1 The manuscript does not include sufficient negative controls, such as untransformed yeast strains exposed to identical conditions. This limits the ability to attribute observed effects directly to genetic modifications.
#2 Statistical analysis is absent for key data points. For instance, growth curves and enzymatic activity results lack standard statistical tests (e.g., ANOVA, t-tests) to confirm significance, reducing the reliability of findings.
#3 The stability of the plasmids in transformed yeast cells under fluoride resistance conditions is not demonstrated. This raises questions about long-term genetic stability during growth assays.
#4 The rationale for using the GAL1 promoter in inducible systems is not adequately explained. Alternative promoters with different strengths and specificities could influence the observed phenotypes.
#5 The manuscript does not provide an ecological or physiological basis for selecting specific fluoride concentrations (e.g., 75 mM NaF) in experiments. This creates uncertainty about the real-world applicability of the findings.
#6 Fluoride resistance is assessed solely through growth assays. Complementary analyses, such as membrane integrity, metabolic changes, or stress marker expression, are necessary to confirm resistance mechanisms.
#7 The authors do not include qPCR or RNA-seq data to verify the expression levels of IPP1 or other introduced genes. Without these, claims of overexpression remain speculative.
#8 The interaction between fluoride and IPP1 is only hypothesized based on prior studies. Direct evidence, such as structural or binding assays, is required to substantiate these claims.
#9 Although the study discusses potential implications for other eukaryotic systems, such as human cells, no experimental data support these assertions.
#10 The study does not explore how modified pyrophosphate metabolism impacts other cellular processes, which could compromise the yeast's overall fitness.
#11 The manuscript does not discuss how these fluoride-resistant strains might behave in natural or industrial settings where fluoride contamination occurs.
#12 The role of pH in modulating fluoride uptake and toxicity is mentioned but not investigated systematically. This oversight weakens the proposed mechanism of HF-mediated fluoride entry.
#13 While the study suggests IPP1 is a major target for fluoride, no biochemical or structural experiments (e.g., crystallography or molecular docking) validate this interaction.
#14 Plasmids that failed to confer resistance are not thoroughly discussed. Exploring these cases could provide insights into unsuccessful resistance mechanisms.
#15 The authors do not define the maximum fluoride concentration that the modified yeast can tolerate, which is crucial for evaluating the efficacy of the modifications.
#16 Details regarding the efficiency and reproducibility of the transformation method are lacking. This limits the applicability of the proposed approach.
#17 The media composition and pH were not systematically optimized for fluoride resistance assays, potentially confounding the results.
#18 The proposed selection strategy for transformants using fluoride resistance lacks comparison with traditional methods, making it challenging to assess its advantages.
#19 Despite referencing similarities between S. cerevisiae and human pyrophosphatases, no experimental data or comparative analysis validate these claims.
#20 The study does not consider alternative fluoride resistance pathways (e.g., fluoride extrusion proteins), which might complement or compete with the described strategy.
#21 The manuscript does not investigate how resistance or enzyme activity evolves over time under sustained fluoride exposure.
#22 Fluorescence microscopy images lack quantitative analysis to confirm the intracellular localization of the introduced proteins, undermining claims about functionality.
#23 The study assumes NaF is representative of all fluoride sources without exploring the effects of other fluoride salts, such as KF or CaF2, on yeast growth and metabolism.
#24 While discussing potential S-phase arrest in some contexts, no cell cycle analysis (e.g., flow cytometry) was performed to validate this hypothesis.
#25 Possible interactions between native and heterologous PPases are not investigated, leaving the compatibility of these systems unaddressed.
#26 The metabolic cost of maintaining elevated PPase activity under fluoride stress is not analyzed, which could impact the fitness of the engineered strains.
#27 Fluorescence microscopy results are qualitative and lack quantitative metrics (e.g., intensity measurements) to support conclusions about protein expression and localization.
#28 Comparative analysis with previous work using other fluoride-resistant systems is sparse, reducing the context and novelty of the study.
#29 The study does not address potential industrial applications for fluoride-resistant yeast strains, such as in biotechnology or environmental remediation.
Minor Concerns
#1 The abstract is overly verbose and contains redundant phrases. Please check your abstract.
#2 The introduction section lacks a clear hypothesis statement.
#3 Some sections of the discussion repeat points from the results section, e.g., Results Section: "Internal levels of PPi significantly increased in soluble extracts of W303-1A cells and YPC3 cells transformed with plasmid pIPP1-416 after growing for only 2 hours in the presence of NaF."; Discussion Section: "In the presence of NaF, intracellular PPi concentrations of YPC3 cells transformed with plasmid pIPP1-416 were higher than those found in cells overexpressing family I sPPases or fluoride-insensitive SPP2."; Results Section: "Transformants recovered the capacity to grow under fermentative conditions and could also grow in the presence of up to 75 mM NaF."; Discussion Section: "YPC3 cells transformed with plasmids bearing genes coding for fluoride-resistant PPases grew in the presence of significantly higher fluoride concentrations than the parental strain W303-1A." Please check all repetition phrases in your manuscript accordingly.
#4 The conclusion does not summarize key findings effectively. Please revise your conclusion.
#5 The authors do not mention limitations of the study.
#6 The paper lacks of emphasis on how results advance the field.
Author Response
General comment for referee 2
First of all, we want to apologize for the misleading writing in some crucial parts of the manuscript (including the Abstract) and thank referee 2 for his/her useful comments and criticisms. We believe they will definitely help us to elaborate an enhanced version of our manuscript.
The general feeling we got from these comments is that there are two fundamental issues that we have not made clear enough in our manuscript:
1- The characteristics of the biological tool we have used to obtain the results described in the manuscript: the engineered yeast strain YPC3. In order to clarify this particular point we have prepared a table that summarizes the most important aspects of this mutant:
Features of YPC3 |
|
Physiological and Metabolic Implications |
Derived from haploid strain W303-1A by substituting the galactokinase (GAL1) promoter for the original IPP1 promoter
|
|
Widespread genetic background, easy to transform with E.coli/S.cerevisiae shuttle plasmids
|
The essential IPP1 gene, that codes for the nucleocytosolic PPase, is only expressed in galactose
|
|
YPC3 grows as well as W303-1A in galactose |
IPP1 "switched off" in glucose -fermenting cells- |
|
6 hours after being transferred from galactose to glucose:
- Growth stops - PPase activity and IPP1 polypeptide reach negligible levels. - Intracellular PPi accumulates
|
IPP1 "switched off" in glycerol -respiring cells- |
|
24 hours after being transferred from galactose to glucose:
- Growth stops - PPase activity and IPP1 polypeptide reach negligible levels. - Intracellular PPi accumulates
|
Can be transformed with high-copy or low-copy plasmids bearing soluble or membrane-bound PPases
|
|
Regain growth on glucose and glycerol |
Remark: YPC3 allows to effectively substitute different PPi-hydrolyzing enzymes for IPP1 in a useful eukaryotic model system such as yeast; in other words, YPC3 permits to test different types of PPases in vivo within the same cellular context.
Relevant references related to YPC3 (all included in manuscript):
- Description of the generation of YPC3: Drake, R.; Serrano, A.; Pérez-Castiñeira, J.R. Biochem. J. 2010, 426, 147–157. doi:10.1042/BJ20091491.
- Description of the consequences of cellular PPi accumulation and kinetics of IPP1 depletion over time under fermenting and respiratory conditions in yeast: Serrano-Bueno, G.; Hernández, A.; López-Lluch, G.; Pérez-Castiñeira, J.R.; Navas, P.; Serrano, A. J. Biol. Chem. 2013, 288, 13082–13092, doi:10.1074/jbc.M112.439349.
- Further characterization of YPC3 and description of some of the plasmids used in this manuscript: Serrano-Bueno, G.; Madroñal, J.M.; Manzano-López, J.; Muñiz, M.; Pérez-Castiñeira, J.R.; Hernández, A.; Serrano, A. Biochim. Biophys. Acta Mol. Cell Res. 2019, 1866, 1019–1033, doi:10.1016/j.bbamcr.2019.02.015.
Pérez-Castiñeira, J.R.; Serrano, A. Front. Plant Sci. 2020, 11, 1240, doi:10.3389/fpls.2020.01240. |
2- The second point that requires clarification is that we do not claim in our manuscript that IPP1 is directly responsible for providing resistance to fluoride or that is an intrinsic part of the mechanisms of fluoride resistance. Our conclusion is that maintaining adequate levels of PPase activity in the cytosol helps yeast cells respond to stress conditions such as that caused by the presence of fluoride in the environment. Our group has been working for nearly three decades in the field of inorganic pyrophosphatases and we have always supported the idea that in certain organisms (such as bacteria, yeast or human cells) these enzymes are housekeeping systems involved in a wide variety of cellular processes due to its essential role in vectorializing metabolism. The latter has been recently discussed in detail [1]
We have re-written more carefully some paragraphs of the manuscript hoping that this will help clarify the major points we want to make.
[1] Wimmer, J. L., Kleinermanns, K., & Martin, W. F. (2021). Frontiers in microbiology, 12, 759359. https://doi.org/10.3389/fmicb.2021.759359 ).
Major Concerns
#1 The manuscript does not include sufficient negative controls, such as untransformed yeast strains exposed to identical conditions. This limits the ability to attribute observed effects directly to genetic modifications.
In our view, the only untransformed yeast strain we could use as a proper control was YPC3 parental strain W303-1A, because untransformed YPC3 does not grow on glucose at all (references 23, 24, 27, 28, 29 of manuscript). The latter is also the case with YPC3 cells transformed with control plasmid pRS699b (Figure 1). In order to compare untransformed YPC3 cells with transformed YPC3 cells we would have to do our experiments in galactose-containing medium (YPGal); however, yeast metabolism is very different in galactose with respect to glucose (the latter being the preferred -and considered the "standard"- carbon source for this organism). This situation poses at least two major problems:
(a) Comparing untransformed YPC3 cells (that have to be grown in galactose) with transformed YPC3 grown in glucose means that we would be comparing cells with very different metabolic programs.
(b) If we had done our experiments in YPGal, transformed YPC3 cells would have expressed two sets of PPases, one encoded by the chromosomal IPP1 gene (controlled by the GAL1 promoter) and another set encoded by the plasmid-borne genes (controlled by constitutive promoters). The latter is exactly the situation we wanted to avoid from the beginning and that is the reason why we have used YPC3 (see the first paragraphs of the Discussion).
#2 Statistical analysis is absent for key data points. For instance, growth curves and enzymatic activity results lack standard statistical tests (e.g., ANOVA, t-tests) to confirm significance, reducing the reliability of findings.
Thanks for the comment, we simply forgot to do it. This has been amended in version 2 of the manuscript
#3 The stability of the plasmids in transformed yeast cells under fluoride resistance conditions is not demonstrated. This raises questions about long-term genetic stability during growth assays.
We can do that by checking the levels of the plasmids and PPase activity after a few days growing cells in the presence of NaF. We would need several weeks to do those experiments. In any case, it must be borne in mind that when grown in the presence of fluoride there are two selection pressures in our system: (a) the presence of glucose or glycerol, that forces YPC3 cells to keep the plasmids bearing PPases (otherwise their growth stop), and (b) fluoride, that inhibits IPP1 thereby lowering the levels of nucleosytosolic PPase activity below the threshold necessary to sustain growth. This may complicate the interpretation of the results.
#4 The rationale for using the GAL1 promoter in inducible systems is not adequately explained. Alternative promoters with different strengths and specificities could influence the observed phenotypes.
We are not using GAL1 promoter to induce IPP1 but to "switch off" this gene (see Table provided). YPC3 mutant was generated by following a classical approach in yeast [2] and has been used in our lab for more than two decades (references 23, 24, 27, 28, 29 of manuscript). This mutant is just a biological tool that allows us to replace IPP1 with different plasmid-borne PPase-encoding genes, making it possible to compare in vivo PPases that are structurally and catalytically very different. In our plasmids the promoter is always that of IPP1 gene, except in plasmid pPPA1-426 (generously donated by Dr. Hernández) where PPA1 gene is under the control of the GPD promoter. The latter situation and its possible implications for our work are discussed in the main text.
[2] Cid A, Perona R, Serrano R. Curr Genet. 1987;12(2):105-10.
#5 The manuscript does not provide an ecological or physiological basis for selecting specific fluoride concentrations (e.g., 75 mM NaF) in experiments. This creates uncertainty about the real-world applicability of the findings.
There is no ecological or physiological basis for choosing these concentrations, they are entirely supported by empirical laboratory studies: we compared our cells with the parental strain W303-1A that happened to tolerate up to 50 mM NaF at pH 4.5 in YPD; then, we increased NaF concentrations (and, very importantly, changed pH values) until we observed a complete growth stop. This happened above 75 mM at pH 4.5. It must be borne in mind that YPD is a rich complex medium (see composition in Materials and Methods) with a high concentration of glucose (2% p/v), moreover, a temperature of 28-30ºC is always maintained along the experiments. Furthermore, plates are autoclaved to remove any other organism that may grow along our strain potentially interfering with our experiments. This scenario is not likely to be confronted by yeast cells in natural environments, therefore, a real-world applicability of these results will require much work and, presumably, also industrial strains. We are just proposing a feasible strategy of obtaining fluoride-resistant/tolerant yeasts.
The concentrations of NaF used in our experiments are very similar to those previously reported in toxicological studies with "wild type-like" strains of S. cerevisiae [3,4]
[3] Johnston, N. R., & Strobel, S. A. (2019). Chemical research in toxicology, 32(11), 2305-2319.
[4] Yoo, J. I., Seppälä, S., & OʼMalley, M. A. (2020). Nature Communications, 11(1), 5459.
#6 Fluoride resistance is assessed solely through growth assays. Complementary analyses, such as membrane integrity, metabolic changes, or stress marker expression, are necessary to confirm resistance mechanisms.
We are not describing molecular mechanisms of resistance but a more general effect: how maintaining sufficient levels of cytosolic PPase activity can help to enhance yeast cells resistance to external fluoride. Our strain can be a valuable tool (they are freely available) to any group that would like to deepen into the mechanisms of fluoride tolerance using "wild type-like" cells that could complement studies done with fluoride-sensitive mutants (see papers by Professor Strobel's group mentioned below).
#7 The authors do not include qPCR or RNA-seq data to verify the expression levels of IPP1 or other introduced genes. Without these, claims of overexpression remain speculative.
We think that showing the levels of PPase activities, their sensitivity to fluoride in vitro, and the immunodetection of family I PPases is enough to sustain these claims in the case of family I PPases (IPP1 from yeast and PPA1 from H. sapiens). We have not claimed overexpression in the case of family II PPase (SPP2 from S. mutans) because we have found lower levels of PPase activity in these cases than those obtained in the parental strain or in YPC3 mutant transformed with plasmids bearing genes encoding family I PPases. Actually, the results with SPP2 are extremely important because it reinforces our point: we do not need very high levels of fluoride-sensitive PPase activity to gain tolerance to this anion, low (even very low) levels of fluoride-insensitive activity being just enough. In other words, resistance to fluoride in yeast requires a minimum of PPase activity regardless of the enzymatic system that accomplishes this task.
Besides the point made in the previous paragraph, after more than 30 years expressing soluble, membrane-embedded and intrinsically disordered proteins in S. cerevisiae we have concluded that levels of transcription do not necessary correlate with levels of translation (or indeed with levels of activity in the case of enzymes), especially in the case of heterologous proteins.
#8 The interaction between fluoride and IPP1 is only hypothesized based on prior studies. Direct evidence, such as structural or binding assays, is required to substantiate these claims.
The work by Professors Rejo Lahti's and Alexander Baykov's groups along the decades clearly established the role of fluoride as a major inhibitor of family I PPases (such as IPP1) [5-10]. Within the "inorganic pyrophosphatases community", fluoride is accepted as a specific inhibitor of family I PPases ("Sodium fluoride is the main inhibitor of family I enzymes" [11]).
[5] A.A. Baykov, J.J. Tam-Villoslado, S.M. Avaeva, Biochimica et Biophysica Acta (BBA) - Enzymology, Volume 569, Issue 2, 1979, Pages 228-238, ISSN 0005-2744, https://doi.org/10.1016/0005-2744(79)90058-5.
[6] Baykov AA, Fabrichniy IP, Pohjanjoki P, Zyryanov AB, Lahti R. Biochemistry. 2000 Oct 3;39(39):11939-47. doi: 10.1021/bi000627u. PMID: 11009607.
[7] Pohjanjoki, P., Fabrichniy, I. P., Kasho, V. N., Cooperman, B. S., Goldman, A., Baykov, A. A., & Lahti, R. (2001). Journal of Biological Chemistry, 276(1), 434-441.DOI: 10.1074/jbc.M007360200
[8] P. Heikinheimo, V. Tuominen, A. Ahonen, A. Teplyakov, B.S. Cooperman, A.A. Baykov, R. Lahti, A. Goldman. Proc. Natl. Acad. Sci. U.S.A. 98 (6) 3121-3126, https://doi.org/10.1073/pnas.061612498 (2001).
[9] Oksanen, E., Ahonen, A. K., Tuominen, H., Tuominen, V., Lahti, R., Goldman, A., & Heikinheimo, P. (2007). Biochemistry, 46(5), 1228-1239. https://doi.org/10.1021/bi0619977.
[10] Baykov, A.A., Anashkin, V.A., Salminen, A. and Lahti, R. (2017). FEBS Lett, 591: 3225-3234. https://doi.org/10.1002/1873-3468.12877
[11] García-Contreras, R., de la Mora, J., Mora-Montes, H. M., Martínez-Álvarez, J. A., Vicente-Gómez, M., Padilla-Vaca, F., ... & Franco, B. (2024). PeerJ, 12, e17496.
#9 Although the study discusses potential implications for other eukaryotic systems, such as human cells, no experimental data support these assertions.
In the Discussion we have used words like "speculate", "suggest" and "hypothesize" when referring to the possible implications of our work for human cells; that is, we are just suggesting some ideas to other groups that may be interested in PPi metabolism and have the expertise and facilities to work with human cells (not our case). We have also speculated with possible implications of our results to bacteria involved in tooth decay.
#10 The study does not explore how modified pyrophosphate metabolism impacts other cellular processes, which could compromise the yeast's overall fitness.
We are describing an effect. Studying these processes will require plenty of work and financial support which we do not have at the moment for this kind of project.
#11 The manuscript does not discuss how these fluoride-resistant strains might behave in natural or industrial settings where fluoride contamination occurs.
We are using lab strains in a laboratory environment under controlled conditions. We do not think that our haploid yeast strains would survive for a long time in natural or industrial settings; therefore, behaviour in fluoride-contaminated natural or industrial settings would require the use of industrial yeasts, whose genetic manipulation is beyond our capabilities and expertise. The conditions in which our strains have been tested are standard in laboratories working with yeast.
#12 The role of pH in modulating fluoride uptake and toxicity is mentioned but not investigated systematically. This oversight weakens the proposed mechanism of HF-mediated fluoride entry.
We have not proposed anything in this respect, this has been published previously and our results (Figures 1 and 3) support these claims. We reproduce a paragraph from a previous publication: "Cellular sensitivity to fluoride depends on the external pH. Fluoride has the highest pKa of any halide at 3.2. Consequently, it is the only halide that remains protonated in mildly acidic environments. As extracellular environments become more acidic, a greater fraction of fluoride forms hydrofluoric acid (HF). Given that HF is uncharged, it can readily diffuse through the lipid bilayer and into the cytoplasm of a cell. The cytoplasm of cells is typically at neutral pH. In this state, the equilibrium shifts to the charged form F–, which cannot pass back through the lipid bilayer. The net effect of exposure of cells to fluoride in acidic environments is the accumulation of fluoride intracellularly, causing downstream toxicity." [12]
The mechanism of fluoride entrance within the cell as HF has also been mentioned elsewhere [13,14].
[12] Johnston NR, Cline G, Strobel SA. Chem Res Toxicol. 2022 Nov 21;35(11):2085-2096. doi: 10.1021/acs.chemrestox.2c00222. Epub 2022 Oct 25.
[13] Marquis, R. E., Clock, S. A., & Mota-Meira, M. (2003). FEMS microbiology reviews, 26(5), 493-510.
[14] Ji, C., Stockbridge, R. B., & Miller, C. (2014). Journal of General Physiology, 144(3), 257-261.
#13 While the study suggests IPP1 is a major target for fluoride, no biochemical or structural experiments (e.g., crystallography or molecular docking) validate this interaction.
The 3D structures of the wild type and seven active site variants of yeast inorganic pyrophosphatase (IPP1), including the fluoride-inhibited enzyme, have been previously determined. The latter has been deposited in the Protein Data Bank [PDB entry 1E6A] released: 19 Mar 2001 (https://www.ebi.ac.uk/pdbe/entry/pdb/1e6a/index). Relevant information can also be found in the following references:
[15] P. Heikinheimo, V. Tuominen, A. Ahonen, A. Teplyakov, B.S. Cooperman, A.A. Baykov, R. Lahti, A. Goldman. Proc. Natl. Acad. Sci. U.S.A. 98 (6) 3121-3126.
[16] Oksanen, E., Ahonen, A. K., Tuominen, H., Tuominen, V., Lahti, R., Goldman, A., & Heikinheimo, P. (2007). Biochemistry, 46(5), 1228-1239.
#14 Plasmids that failed to confer resistance are not thoroughly discussed. Exploring these cases could provide insights into unsuccessful resistance mechanisms.
All plasmids tested were able to confer resistance to fluoride at different degrees when compared with the parental strain. The only one that failed to do so was pPPA1-426 under respiratory conditions (Figure 3); however, this construction struggled to support growth also in the absence of fluoride. This is discussed in lines 424-445 of the first manuscript (6th paragraph of the Discussion). Plasmid pRS699b is just an "empty plasmid", that is, it bears no gene after the IPP1 promoter; therefore, it is used as a negative control.
#15 The authors do not define the maximum fluoride concentration that the modified yeast can tolerate, which is crucial for evaluating the efficacy of the modifications.
Yeast cells struggle to grow at NaF concentrations higher than 75 mM (pH 4.5). We thought that Figure 1 had plenty of information and did not want to make it more complicated by inserting another column with photographs without cells. We have added a phrase in the legend to the figure concerning this point.
#16 Details regarding the efficiency and reproducibility of the transformation method are lacking. This limits the applicability of the proposed approach.
We have to apologize again for making a gross mistake in the abstract: we are not describing a transformation method but a selection method for the transformants. YPC3 cells were transformed by using the method of Schiestl and Gietz [reference 36 of the manuscript], which is a highly efficient method to transform yeast cells commonly used by most yeast laboratories. We have re-written the text in Materials and Methods to make it clearer.
#17 The media composition and pH were not systematically optimized for fluoride resistance assays, potentially confounding the results.
We use YPD, the most standard culture medium for yeast, and we have compared growth of the same mutant transformed with a series of plasmids under exactly the same culture conditions. In this respect, previous work on fluoride toxicity in yeast has been done on YPD [17-19]. The use of this culture medium further guarantees that any group with a basic expertise in yeast can use our strains and reproduce our results.
[17] Johnston, N. R., & Strobel, S. A. (2019). Chemical research in toxicology, 32(11), 2305-2319.
[18] Yoo, J. I., Seppälä, S., & OʼMalley, M. A. (2020). Nature Communications, 11(1), 5459.
[19] Johnston, N. R., Cline, G., & Strobel, S. A. (2022). Chemical Research in Toxicology, 35(11), 2085-2096.
#18 The proposed selection strategy for transformants using fluoride resistance lacks comparison with traditional methods, making it challenging to assess its advantages.
The selection procedure does not require fluoride, it does require glucose (or glycerol, if respiring conditions are used, although this is quite unusual in yeast). Advantages are described in the main text: the use of a standard, cheap and easy-to-make medium that only requires yeast extract, peptone and glucose to be prepared. These reagents are normally present in laboratories involved in Microbiology, Biochemistry and/or Molecular Biology. There is no need to make more complicated and expensive synthetic media that require pure aminoacids and nucleotides; moreover, it is a selection procedure not based on auxotrophy (which might be used for additional genetic manipulations in the case of W303-1A or YPC3).
The use of fluoride is optional, as we describe in the manuscript, it is useful to double-check the transformants: in our tests, all the colonies selected from the plates shown in Figure 7 could be subsequently grown on pH 5-buffered YPD plates supplemented with 70-75 mM NaF. Fluoride is also cheaper and more stable than other chemicals used for selection of yeast transformants such as G418, nourseothricin, hygromycin B, or chloramphenicol [20].
[20] Stepchenkova EI, Zadorsky SP, Shumega AR, Aksenova AY. Int J Mol Sci. 2023 Jul 26;24(15):11960.
#19 Despite referencing similarities between S. cerevisiae and human pyrophosphatases, no experimental data or comparative analysis validate these claims.
Human PPA1 has been crystallized and its 3D structure compared with that of IPP1 [21]. In this paper the following paragraph can be read:
"Among the sPPases with known structures, human PPA1 share the highest sequence homology with the sPPases from S. japonicum and S. cerevisiae (about 70% similarity, see Figure 1). The structures of the sPPases from these three species are similar. Superimposition of the monomeric human PPA1 structure to the monomeric structure of sPPase from S. japonicum (PDB code 4QLZ) and S. cerevisiae (PDB code 2IHP) gave a RMSD of 0.98 Å (based on 3032 common atoms) and 0.97 Å (based on 2981 common atoms) respectively. Figure 2B shows a superimposition of the monomeric structures of human PPA1 and Y-PPase. The two structures superimpose well in most parts of the molecules, including the active site"
In the same paper there is this paragraph:
"The human PPA1 structure does not contain the substrate at the putative active site. To gain insights into how the speculative active site residues (inferred from structure-based sequence alignment with Y-PPase) orient in relative to the pyrophosphate substrate, the substrate was modeled into the human PPA1 structure by superimposition of the structure with a substrate-bound, fluoride-inhibited Y-PPase structure (PDB code 1RE6A).47 The overall structures superimposed well with a small RMSD of 0.98 Å (Figure 2B). " (Comment: underlining is ours)
[21] Niu, H., Zhu, J., Qu, Q., Zhou, X., Huang, X., & Du, Z. (2021). Proteins: Structure, Function, and Bioinformatics, 89(7), 853-865.
#20 The study does not consider alternative fluoride resistance pathways (e.g., fluoride extrusion proteins), which might complement or compete with the described strategy.
Definitely, we have not been clear enough about our conclusions. We do not say categorically that that IPP1 is directly involved in the mechanism of fluoride resistance. At the end of the 12th paragraph (lines 512-522 of the first version of the manuscript) we say:
"This evidence supports previous reports showing that yeast cells have alternative ways to develop resistance to fluoride [7,42] and raises the question of what role IPP1 actually plays in the mechanism of fluoride resistance. PPi hydrolysis has been suggested to be "an ancient mechanism that imparts irreversibility (...) functioning like a ratchet’s pawl to vectorialize the life process toward growth" [52]. This means that increasing levels of fluoride in the cytosol would gradually inhibit multiple enzymatic systems, including IPP1 itself, which in turn would slow down anabolism hindering the capacity of the cell to respond. Overexpression of IPP1 (or expression of a fluoride-resistant PPase) might prevent, or at least delay, the anabolic collapse thus allowing the cell a precious time to activate the different mechanisms that help cope with fluoride toxicity [7]."
#21 The manuscript does not investigate how resistance or enzyme activity evolves over time under sustained fluoride exposure.
See our answer to point #3
#22 Fluorescence microscopy images lack quantitative analysis to confirm the intracellular localization of the introduced proteins, undermining claims about functionality.
They are definitely non-thorough qualitative analyses, we are not claiming otherwise. We just wanted to demonstrate that the yeast transformants selected with our method could be easily double-checked (or triple-checked if we consider our answer to point #18) by direct visualization using fluorescence microscopy, as long as plasmids piGMVP-416 and pIPP1GFP-699 are used for transformation. We decided to show an ample microscopy field (40x) to show that fluorescence appeared in many cells. Similar more detailed images than those shown here have been previously published by our group [22,23]:
[22] Serrano-Bueno, G., Madroñal, J. M., Manzano-López, J., Muñiz, M., Pérez-Castiñeira, J. R., Hernández, A., & Serrano, A. (2019). Biochimica et Biophysica Acta (BBA)-Molecular Cell Research, 1866(6), 1019-1033.
[23] Pérez-Castiñeira, J.R.; Serrano, A. Front. Plant Sci. 2020, 11, 1240. (In supplementary figures)
#23 The study assumes NaF is representative of all fluoride sources without exploring the effects of other fluoride salts, such as KF or CaF2, on yeast growth and metabolism.
We mention (lines 257-259 of the first manuscript) that results obtained with KF were identical to those obtained with NaF. CaF2 is quite insoluble in aqueous media, which would make very difficult to reach the fluoride concentrations needed to observe effects on our strains (above 50 mM) by using this salt (not to mention the effects that high concentrations of calcium could exert on the observed phenotypes). In connection with this, we added a magnesium salt (MgCl2 or MgSO4) to our YPD plates supplemented with NaF in order to check for a possible protective effect of Mg2+ on our cells (as we had previously seen for salt stress in [24]) but a conspicuous white precipitate of MgF2 formed in the plates at all pH values used for our experiments. All cells (including parental strain W303-1A) grew normally in these plates suggesting that free fluoride concentration had significantly decreased after addition of Mg2+ salts. CaF2 and MgF2 have similar solubilities in aqueous media.
NaF is the fluoride salt utilized in all previous studies of fluoride toxicity in yeast (see references in our answer to point #17)
[24] Pérez-Castiñeira, J.R.; Serrano, A. Front. Plant Sci. 2020, 11, 1240.
#24 While discussing potential S-phase arrest in some contexts, no cell cycle analysis (e.g., flow cytometry) was performed to validate this hypothesis.
This information has been previously published by our group:
[25] Serrano-Bueno, Gloria, Hernández, Agustín, López-Lluch, Guillermo, Pérez-Castiñeira, José Román, Navas, Plácido, Serrano, Aurelio. 2013. Journal of Biological Chemistry. 288 (18). 13082-13092.
[26] Hernández Agustín , Herrera-Palau Rosana , Madroñal Juan M. , Albi Tomás , López-Lluch Guillermo, Perez-Castiñeira José R. , Navas Plácido , Valverde Federico , Serrano Aurelio. Frontiers in Plant Science. VOL.7 (2016).
#25 Possible interactions between native and heterologous PPases are not investigated, leaving the compatibility of these systems unaddressed.
The advantage of mutant YPC3 is that native PPase (IPP1) is undetectable (activity and polypeptide) when grown on YPD or YPGly, as shown in this manuscript and others previously published by our group.
#26 The metabolic cost of maintaining elevated PPase activity under fluoride stress is not analyzed, which could impact the fitness of the engineered strains.
We think that studying the fitness of the engineered strains is beyond the scope of this manuscript
#27 Fluorescence microscopy results are qualitative and lack quantitative metrics (e.g., intensity measurements) to support conclusions about protein expression and localization.
See our answer to point #22
#28 Comparative analysis with previous work using other fluoride-resistant systems is sparse, reducing the context and novelty of the study.
We have compared our results with those obtained by Professor Scott A. Strobel’s group at Yale University (https://strobel.yale.edu/)[27,28]. These are the only previous studies similar to those described here that we know of; actually, in a review published in 2024 [29] only the articles by Prof. Strobel's group were cited when discussing fluoride resistance in eukaryotic microbial cells. It would be great for us if referee 2 can suggest previous work that can help us to increase the context of our study.
The novelty of our study lies on the use of a very useful biological tool (mutant YPC3) to study the involvement of PPi metabolism in fluoride resistance in a "wild-type" yeast. In previous reports, fluoride-sensitive mutants were used; these were generated by disruption of FEX1 and FEX2, the genes that code for the yeast homologs of bacterial fluoride exporter Fluc [30,31]. We think that the studies on the effects exerted by fluoride on a biological system like YPC3 can complement those obtained with fluoride-sensitive mutants.
[27] Johnston, N.R.; Cline, G.; Strobel, S.A. Chem. Res. Toxicol. 2022, 35, 2085–2096, doi:10.1021/acs.chemrestox.2c00222.
[28] Johnston, N. R. & Strobel, S. A. Chem. Res. Toxicol. 32, 2305–2319 (2019)].
[29] Stockbridge, R.B.; Wackett, L.P. Nat. Commun. 2024, 15, 4593, doi:10.1038/s41467-024-49018-1)
[30] Li, S., Smith, K. D., Davis, J. H., Gordon, P. B., Breaker, R. R., & Strobel, S. A. (2013). Proceedings of the National Academy of Sciences, 110(47), 19018-19023.
[31] Yoo, J. I., Seppälä, S., & OʼMalley, M. A. (2020). Nature Communications, 11(1), 5459.
#29 The study does not address potential industrial applications for fluoride-resistant yeast strains, such as in biotechnology or environmental remediation.
We are not experts in the fields of biotechnology applications or environmental remediation, but biochemists/molecular biologists that have worked on the molecular-physiological relevance of inorganic pyrophosphatases (both soluble and membrane-bound) for nearly 30 years. We hope that, if this paper is finally published, other groups can implement practical applications based on our findings.
Minor Concerns
#1 The abstract is overly verbose and contains redundant phrases. Please check your abstract.
#2 The introduction section lacks a clear hypothesis statement.
#3 Some sections of the discussion repeat points from the results section, e.g., Results Section: "Internal levels of PPi significantly increased in soluble extracts of W303-1A cells and YPC3 cells transformed with plasmid pIPP1-416 after growing for only 2 hours in the presence of NaF."; Discussion Section: "In the presence of NaF, intracellular PPi concentrations of YPC3 cells transformed with plasmid pIPP1-416 were higher than those found in cells overexpressing family I sPPases or fluoride-insensitive SPP2."; Results Section: "Transformants recovered the capacity to grow under fermentative conditions and could also grow in the presence of up to 75 mM NaF."; Discussion Section: "YPC3 cells transformed with plasmids bearing genes coding for fluoride-resistant PPases grew in the presence of significantly higher fluoride concentrations than the parental strain W303-1A." Please check all repetition phrases in your manuscript accordingly.
#4 The conclusion does not summarize key findings effectively. Please revise your conclusion.
#5 The authors do not mention limitations of the study.
#6 The paper lacks of emphasis on how results advance the field.
Answer: A new throroughly revised version of the manuscript has been uploaded in the journal system. We hope it will correctly address your concerns
Round 2
Reviewer 2 Report
Comments and Suggestions for Authors
Thank you for your detailed responses and clarifications. I appreciate the effort you have put into addressing the issues raised. Your explanations were clear, and the revisions have adequately resolved the concerns. Considering the improvements made, I am pleased to recommend the acceptance of your manuscript. Congratulations on your work!